# Deep learning-based differential gut flora for prediction of Parkinson's

Bo Yu[1]*, Hang Zhang[1☯]*, Min Zhang[2☯]

1 School of Computer Science and Technology, Harbin University of Science and Technology, Harbin, China.
2 College of Computer and Information Technology, Northeast Petroleum University, China

☯ These authors contributed equally to this work.
* niuchatang@163.com (HZ); yubo@hrbust.edu.cn (BY)

## Abstract

### Background

There had been extensive research on the role of the gut microbiota in human health and disease. Increasing evidence suggested that the gut-brain axis played a crucial role in Parkinson's disease, with changes in the gut microbiota speculated to be involved in the pathogenesis of Parkinson's disease or interfere with its treatment. However, studies utilizing deep learning methods to predict Parkinson's disease through the gut microbiota were still limited. Therefore, the goal of this study was to develop an efficient and accurate prediction method based on deep learning by thoroughly analyzing gut microbiota data to achieve the diagnosis of Parkinson's disease.

### Methods

This study proposed a method for predicting Parkinson's disease using differential gut microbiota, named the Parkinson Gut Prediction Method (PGPM). Initially, differential gut microbiota data were extracted from 39 Parkinson's disease (PD) patients and their corresponding 39 healthy spouses. Subsequently, a preprocessing method called CRFS (combined ranking using random forest scores and principal component analysis contributions) was introduced for feature selection. Following this, the proposed LSIM (LSTM-penultimate to SVM Input Method) approach was utilized for classifying Parkinson's patients. Finally, a soft voting mechanism was employed to predict Parkinson's disease patients.

### Results

The research results demonstrated that the Parkinson gut prediction method (PGPM), which utilized differential gut microbiota, performed excellently. The method achieved a mean accuracy (ACC) of 0.85, an area under the curve (AUC) of 0.92, and a receiver operating characteristic (ROC) score of 0.92.

### Conclusion

In summary, this method demonstrated excellent performance in predicting Parkinson's disease, allowing for more accurate predictions of Parkinson's disease.

**Data Availability Statement:** All relevant data are within the manuscript and its Supporting Information files.

**Funding:** The author(s) received no specific funding for this work.

**Competing interests:** The authors have declared that no competing interests exist.

## 1.Contexts

Parkinson's disease [1] is a common multifunctional dysfunction and neurodegenerative disorder among elderly people, and its prevalence is second only to that of Alzheimer's disease [2]. With the increase in population size and the intensification of aging trends, the burden of Parkinson's disease on society and individual health will continue to increase. According to research predictions, by 2040, the global number of diagnosed cases of Parkinson's disease [3] will exceed 10 million.

Research indicates that the gut microbiota interacts with the autonomic and central nervous systems through various pathways, and dysbiosis of the gut microbiota may affect both the enteric nervous system and the central nervous system. Previous studies have revealed the existence of the brain-gut-microbiota axis, where bidirectional interactions between the gut microbiota and the human nervous system could lead to central nervous system diseases. The gut microbiota, also known as the "second brain," can influence brain activity under both physiological and pathological conditions through the gut-microbiota-brain axis. Changes in the gut microbiota have been linked to several psychiatric and neurological diseases, including schizophrenia [4], depression [5,6], and autism [7]. Recently, numerous studies have shown significant differences in the composition of the gut microbiota between Parkinson's disease patients and healthy controls, with metagenomic [8] studies further revealing the correlation between Parkinson's and abnormalities in the gut microbiome. However, research on the use of the gut microbiota as a predictive tool for Parkinson's disease is still relatively scarce. Therefore, exploring a method to predict Parkinson's disease using the gut microbiota is highly important. Therefore, the aim and objective of this study are to develop an efficient and accurate method for predicting Parkinson's disease based on gut microbiota, in order to achieve the diagnosis of Parkinson's disease. By incorporating deep learning technology, we aim to capture subtle differences in gut microbiota to provide new perspectives and tools for predicting Parkinson's disease, thereby offering scientific support for its diagnosis.

The diagnosis of Parkinson's disease relies on core clinical features and follows standard clinical criteria to improve accuracy. For example, the UK Parkinson's Disease Society Brain Bank (UKPDSBB) has established comprehensive standards, including criteria such as bradykinesia and exclusion of other potential causes. However, these standards still have limitations and rely on the expertise of neurologists. With the development of artificial intelligence and the increasing demand for healthcare, AI-based methods have been applied to the automated diagnosis of Parkinson's disease. Common methods, such as EEG [9], gait analysis [10], voice analysis, and brain imaging, use biomarkers of Parkinson's disease for automated detection. Traditional machine learning models need to extract features from biomarkers and select significant features for model training. Although AI-based methods have potential in the automated diagnosis of Parkinson's disease, they have limitations. These methods may be constrained by technical limitations and challenges in data collection during practical applications. Additionally, the accuracy and reliability of biomarkers still have certain limitations. Furthermore, individual differences and the complexity of cases may affect the applicability and generalizability of the models. Moreover, these methods are typically used as auxiliary diagnostic tools and still require the professional judgment and clinical experience of doctors. It is worth noting that there is relatively limited research on the use of the gut microbiota to predict PD. Therefore, this study utilized gut microbiota prediction combined with artificial intelligence methods to predict Parkinson's disease.

In this article, a Parkinson's disease prediction method called Differential Gut Microbiota for Parkinson's Prediction (PGPM), which can predict Parkinson's disease more accurately, is proposed. First, the PGPM method introduces the CRFS preprocessing method for feature

selection, reducing the dimensionality of features; second, PGPM differs from individual classifiers, improving prediction accuracy; and finally, the final prediction result is obtained through soft voting. Under 10-fold cross-validation, PGPM achieves mean ACC, AUC, and ROC values of 0.85, 0.92, and 0.92, respectively, which are significantly higher than those of existing methods.

## 2. Materials and methods

### 2.1 Microbiota datasets

The data for this study were obtained from a cross-sectional study of the gut microbiota of Parkinson's disease patients in the Central China region [11]. The dataset included 39 Parkinson's disease patients (PD) with a BMI of 23.15 kg/m2 and their healthy spouses (SP) with a BMI of 24.22 kg/m2. The diagnosis of Parkinson's disease was based on the 2015 Movement Disorder Society Parkinson's diagnostic criteria, with the core criterion being the presence of Parkinsonian symptoms. If a patient exhibited bradykinesia along with either resting tremor or rigidity, they were considered to have Parkinson's syndrome.

### 2.2 Transcriptome sequencing

The data were collected by sampling the subjects' feces, which were then stored at -80°C. DNA was extracted from the feces using the MetaHIT protocol, and the DNA concentration was estimated using a Qubit instrument. After DNA extraction, gene libraries were prepared according to the manufacturer's instructions and sequenced. The raw sequencing data have been deposited under the accession number PRJNA588035. The quality of the raw metagenomic data was checked using the FastQC tool, followed by trimming low-quality data and removing unwanted genomes. Subsequently, taxonomic analysis was performed, and the read abundance was estimated after processing. The relative abundance was calculated by multiplying the sequence count and rounding the result.

### 2.3 Overall framework of the forecasting methodology

In this study, a method for predicting Parkinson's disease patients using differential microbiota was implemented. Building upon previous research, improvements were made in data preprocessing, specifically in feature selection and dimensionality reduction, and a method combining neural networks and machine learning was developed. The overall framework of the PGPM method constructed in that article was illustrated in Fig 1, which consisted of three modules: the CR (CRFS Preprocessing Layer) layer, the LS (LSTM-SVM Layer) layer, and the OP (Output Layer) layer. The CR layer was responsible for the initial processing and selection of the raw Parkinson's gut microbiota data to meet the network input requirements. The LS layer utilized LSTM and SVM as shown in Fig 1 to construct the network, while the OP layer provided Parkinson's prediction results through soft voting. The training of the PGPM method network employed the Adam optimization algorithm. Unlike traditional methods that used a single classifier for training, the PGPM method significantly improved model performance by not relying on a single classifier.

### 2.4 CRFS preprocessing methods

In previous studies, a single feature selection method was often used. While this approach could yield simplified features and to some extent improve model performance by reducing model complexity, to enhance the reliability of the selection, the PGPM introduced the CRFS data preprocessing method, as illustrated in Fig 2. Unlike previous research, the CRFS data

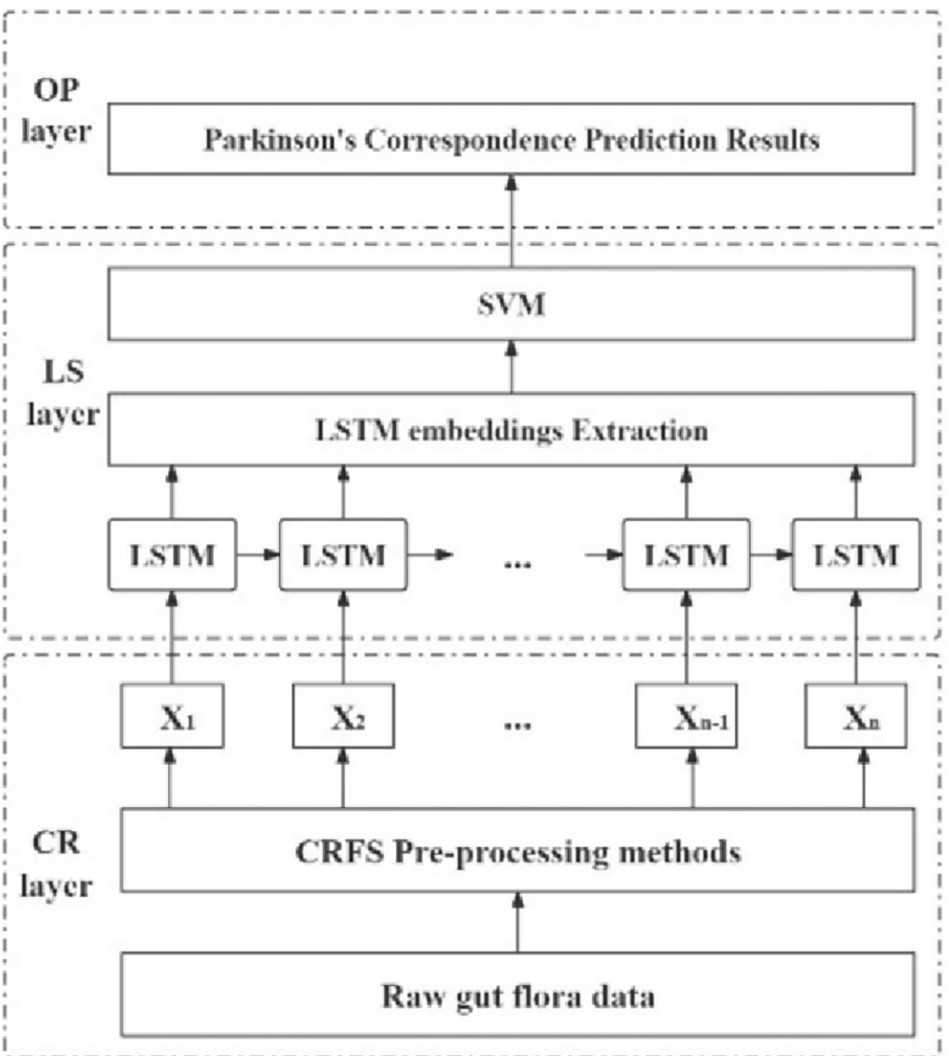

**Fig 1. PGPM framework diagram.**

preprocessing method comprehensively considered the advantages of both random forest (RF) [12] and principal component analysis (PCA) [13] feature dimensionality reduction methods.

Parkinson's gut microbiome data typically contained multiple variables, i.e., different types of microbial populations. One of the advantages of random forest was that it could estimate

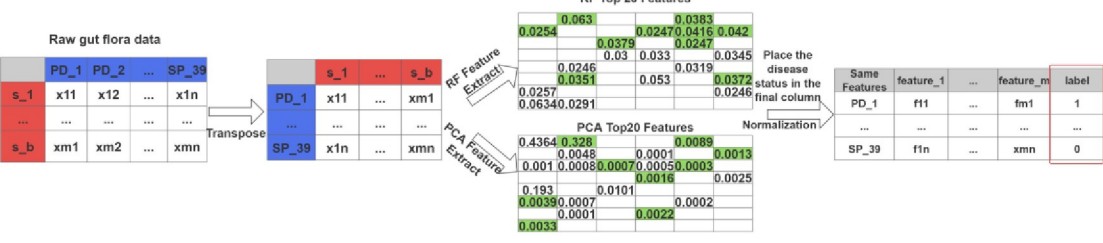

**Fig 2. Diagram of the CRFS preprocessing process.**

the importance of each feature and identify the most important features during classification. By selecting important microbes as inputs, the model complexity could be simplified, computational efficiency could be improved, and overfitting risk could be reduced [14].

On the other hand, principal component analysis (PCA) could be used for dimensionality reduction by explaining most of the variance in the variables with a few principal components. This helped to better understand the data and extract the most informative microbial populations. In PCA, covariance played a crucial role. By calculating the covariance matrix between microbial variables, relationships and correlations could be understood. The covariance matrix could represent the trend of how different microbial populations increased or decreased together. If two microbial populations had high positive covariance, it indicated they had similar patterns of variation in the sample. Conversely, high negative covariance indicated opposite trends in variation. By arranging the covariance matrix according to the size of variance and selecting the top principal components, most of the variance in the data could be explained, achieving dimensionality reduction and retaining the most informative microbial populations. The main method of extracting features was to transform the feature space through the relationships between attributes and map the original feature space to a lower-dimensional feature space, thus accomplishing dimensionality reduction. PCA (Principal Component Analysis) reduced dimensionality through the prior inertia between multidimensional datasets.

The primary method for feature extraction was to transform the feature space by exploring the relationships among attributes and mapping the original feature space into a lower-dimensional feature space to achieve dimensionality reduction. PCA (Principal Component Analysis) achieved dimensionality reduction by leveraging the inertia between multidimensional data groups.

The preprocessing method for feature selection in CRFS involves the following steps:

Step 1: After the distinct gut microbiota associated with Parkinson's disease species are extracted, where the original gut microbiota data in each column represent a sample, the microbiota needs to be transposed. This transformation changes the data so that each row represents a sample, and each column corresponds to a distinct gut microbiota.

Step 2: For the transposed data, feature selection is conducted using two methods: random forest (RF) and principal component analysis (PCA). The distinct gut microbiota were ranked based on importance scores using random forest, and the top 20 were selected. Subsequently, PCA was used to rank the distinct microbiota, selecting the top 20. The shared top 20 microbiota from both methods were chosen as input features. The covariance calculation formula for PCA is shown below (Eq 2–1).

$$cov(X, Y) = \frac{\sum_{i=1}^{n}(X_i - \bar{X})(Y_i - \bar{Y})}{n - 1} \qquad (2-1)$$

Step 3: Extract the corresponding data of the common features from the top 20 features sorted by both methods. The highlighted green portion in Fig 2 represents the identical features.

Step 4: Normalize the extracted data. As the species abundance of the gut microbiota is purely numerical, if the abundance of a certain microorganism is too large, it may lead to an overly significant weight for that microorganism. Therefore, after feature selection and dimensionality reduction, the abundance of each microorganism was normalized to ensure equal weight for each microorganism during the training process, thus ensuring the model's accuracy. The normalization calculation Formula (2–2) is as follows, where *x* represents the original data, *Min* represents the minimum value of the data, *Max* represents the maximum

value of the data, and $x'$ represents the transformed data:

$$x' = \frac{x - Min}{Max - Min} \qquad (2-2)$$

Step 5: Add the corresponding disease status labels to the extracted data after each sample.

## 2.5 LSIM

In that study, the classifier for the PGPM method was based on a combined classification strategy of LSTM-SVM [15]. The structure of the LSIM [16] classifier model was illustrated in Fig 3. By leveraging the advantages of LSTM neural networks in storing long-term information and the generalization and accuracy advantages of SVM in handling classification problems, these two methods were integrated. The LSIM method utilized SVM as the classifier, where the output from the second-to-last layer of LSTM was transformed into the input feature vector for SVM. This approach further involved training SVM using the previous feature vectors, which meant extracting features with LSTM and then classifying them with SVM. The combination of LSTM and SVM not only enhanced the precision and effectiveness of feature extraction but also improved the accuracy of classification results.

The Support Vector Machine (SVM) was a classic machine learning method commonly used for binary classification tasks. Its principle involved constructing an optimal decision hyperplane to separate data samples of different classes. For new input data, classification was determined based on which side of the hyperplane it fell on, thus achieving the classification task. In that study, the SVM utilized the Radial Basis Function (RBF) kernel. The RBF kernel was one of the commonly used kernel functions. It measured the similarity of sample points in a high-dimensional space by calculating the Euclidean distance between the sample points and

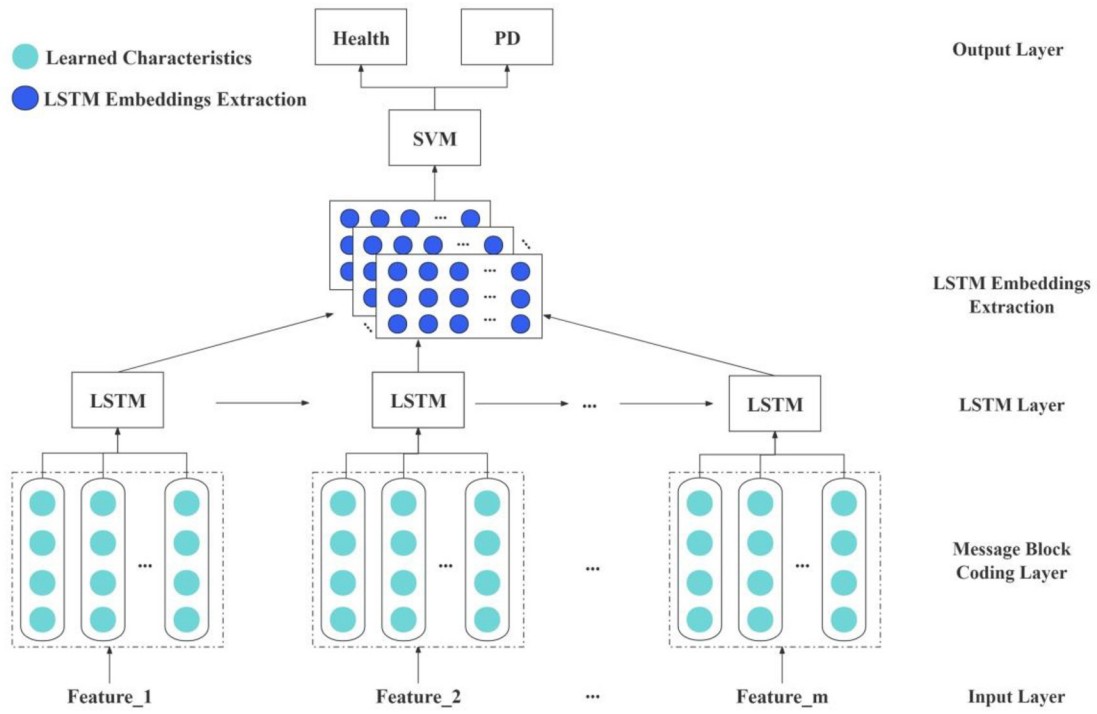

**Fig 3. LSIM structure.**

support vectors. The role of the kernel function in the SVM model was to introduce nonlinear transformations, map the data from the input space to a higher-dimensional feature space, making the data more easily separable in the new feature space. The formula for the RBF kernel function is shown below:

$$K(x, y) = \exp(-\gamma \|x - y\|^2) \tag{2-3}$$

$\gamma$ was a parameter in the RBF kernel function that controlled the rate of decay of the distance between samples, with a larger $\gamma$ causing the similarity between samples to decrease faster, i.e., the similarity between samples that were farther away decreased, and vice versa. Therefore, choosing a suitable $\gamma$ value was highly important for SVM performance and classification results. Too large or too small $\gamma$ values could lead to overfitting or underfitting of the model. In this study, the framework was used to automatically adjust the $\gamma$ values in the framework to adaptively select the appropriate $\gamma$ values. The Formula (2–4) is shown below:

$$\gamma = \frac{1}{n\_features * X.var()} \tag{2-4}$$

Where $n\_features$ denoted the number of features, and $X.var()$ denoted the variance of each feature in the input data X. The method could automatically adjust the input data according to the different scales of its $\gamma$ values to better fit the data.

However, in some tasks, the "sparse" and "discrete" features in the input data made it difficult to detect relationships between data points, which were often crucial for determining the overall relationships in the input. In contrast, Long Short-Term Memory (LSTM) networks could capture dependencies in input information, and were particularly suitable for handling sequential data. LSTMs excelled at handling long-term dependencies and temporal relationships within sequences.

LSTM was a special type of RNN. LSTM introduced the concepts of memory cells, input gates, output gates, and forget gates, enabling it to capture dependencies in input information. The input gate selected relevant information to update the input memory cell. The forget gate determined whether the input and output information should pass through. If the result of the forget gate was close to zero, the information was forgotten, while if it was close to one, the information was retained. This operation at the forget gate allowed LSTM to address the issues of gradient explosion and vanishing gradients. LSTM overcame the short-term memory limitations of RNNs; when a sequence was long, an RNN struggled to propagate information from earlier time steps to later ones, whereas LSTM could learn long-term dependencies, remember information from earlier time steps, and thus establish context.

The LSTM is calculated using the following information:

1. $x_t$: Enter the data at time t.

2. $h_{t-1}$: the hidden state at time t-1.

3. $c_{t-1}$: the state of the cell at time t.

Given $x_t, h_{t-1}2$ and $c_{t-1}$, the LSTM prioritizes the computation of forgetting gates, input gates, output gates and candidate contexts with the Formulas (2–5) to (2–8):

$$f_t = \sigma([x_t; h_{t-1}]W_f + b_f) \tag{2-5}$$

$$i_t = \sigma([x_t; h_{t-1}]W_i + b_i) \tag{2-6}$$

$$o_t = \sigma([x_t; h_{t-1}] W_o + b_o) \qquad (2-7)$$

$$\tilde{c}_t = F_{\tilde{c}}([x_t; h_{t-1}] W_{\tilde{c}} + b_{\tilde{c}}) \qquad (2-8)$$

The LSTM is based on $f_t c_{t-1}$, the $i_t$ and $\tilde{c}_t$ are used to calculate the cell state at the current step $c_t$, as shown in Eq (2–9):

$$c_t = f_t c_{t-1} + i_t \tilde{c}_t \qquad (2-9)$$

LSTM utilizes the $o_t$ and $c_t$ to compute the hidden state of the current step as shown in Eq (2–10):

$$h_t = o_t * F(c_t) \qquad (2-10)$$

Finally, the hidden state $h_t$ is the same as the output given by the LSTM at time t.

LSTM was commonly used for classification tasks, and the softmax layer was a commonly used classification layer for performing binary classification tasks. The output of the softmax layer could be interpreted as the estimated probability of the sample belonging to a certain class. In binary classification tasks, a threshold was applied to convert the probability value into a specific class label. If the probability was greater than the threshold, the sample was predicted to belong to the positive class; otherwise, it was predicted as the negative class. In this experiment, the cross-entropy loss function, which affected the classification layer of LSTM, was used. Therefore, when the features of the data were linearly inseparable, combining SVM with LSTM could address the same classification problem from different perspectives. This combination may have rendered the originally inseparable classification problem linearly separable, thereby further improving the classification performance.

## 3 Experimental results

### 3.1 Network training

This study is implemented based on Python (3.9.12) using publicly available standard libraries: pandas (1.5.2), numpy (1.22.4), scikit-learn (1.2.0), torch (1.12), and matplotlib (3.6.2). To avoid underfitting or overfitting, the DataLoader method is used to randomly shuffle the samples in the dataset at the beginning of each epoch. This helps the model better learn the data distribution and improves its generalization ability.

The network training mainly focuses on the hidden layers. In this study, 10-fold cross-validation is used to evaluate the model's performance. First, the entire dataset is divided into 10 parts, each of which is used as a training set in turn, with the rest used as a test set. Then, the dataset undergoes 10 rounds of training, and during each training loop, an internal epoch is used for multiple rounds of training. The training set is divided into small batches for training, and the model's parameters are updated through backpropagation and the Adam optimizer. After the training is completed, the penultimate layer output of the LSTM is extracted as a feature vector. These feature vectors and the test set are used for training, prediction, and accuracy calculation. Finally, after each round of validation, the accuracy is stored in a list.

This experiment conducts comparative tests on multiple models with the same hyperparameter settings. The specific settings are as follows: the training epoch is 300, the initial learning rate is 0.001 [17], the batch size is set to 6, and the optimization algorithm used is Adaptive Moment Estimation (Adam) [18]. The GPU used for training is an NVIDIA GeForce GTX1060 laptop GPU, with 16GB of memory and 1280 CUDA cores.

**Table 1. CRFS preprocessing results.**

| RF | | Species name | PCA | | Species name |
|---|---|---|---|---|---|
| | 1 | Bilophila_unclassified | | 1 | Rothia_dentocariosa |
| | 2 | **Bifidobacterium_dentium** | | 2 | **Bifidobacterium_dentium** |
| | 3 | Ruminococcaceae_bacterium_D16 | | 3 | Scardovia_inopinata |
| | 4 | **Alistipes_putredinis** | | 4 | Scardovia_unclassified |
| | 5 | **Alistipes_indistinctus** | | 5 | **Scardovia_wiggsiae** |
| | 6 | **Scardovia_wiggsiae** | | 6 | Olsenella_unclassified |
| | 7 | **Gemella_haemolysans** | | 7 | **Bacteroides_coprocola** |
| | 8 | Subdoligranulum_unclassified | | 8 | Bacteroides_sp_3_1_19 |
| | 9 | Peptostreptococcaceae_noname_unclassified | | 9 | Butyricimonas_synergistica |
| | 10 | Clostridium_leptum | | 10 | **Parabacteroides_goldsteinii** |
| | 11 | Clostridium_hathewayi | | 11 | **Alistipes_indistinctus** |
| | 12 | Lachnospiraceae_bacterium_3_1_57FAA_CT1 | | 12 | **Alistipes_putredinis** |
| | 13 | Clostridium_citroniae | | 13 | Alistipes_sp_AP11 |
| | 14 | Bilophila_wadsworthia | | 14 | single_cell_isolate_TM7b |
| | 15 | Subdoligranulum_variabile | | 15 | **Gemella_haemolysans** |
| | 16 | **Bacteroides_coprocola** | | 16 | Lactobacillus_gasseri |
| | 17 | **Parabacteroides_goldsteinii** | | 17 | **Lactobacillus_salivarius** |
| | 18 | **Lactobacillus_salivarius** | | 18 | Leuconostoc_pseudomesenteroides |
| | 19 | Clostridium_symbiosum | | 19 | Streptococcus_pasteurianus |
| | 20 | Oxalobacter_formigenes | | 20 | Clostridium_asparagiforme |

## 3.2 CRFS preprocessing results

To address the issue of redundant information in the data that may lead to suboptimal classification, the CRFS preprocessing method is used to retain relevant information and eliminate irrelevant information. Table 1 presents partial results of feature selection using the CRFS data preprocessing method.

Table 1 shows that, within the CRFS preprocessing method, the Random Forest (RF) and PCA (Principal Component Analysis) methods share 8 identical microbes among their top 20 features, which are highlighted in bold, including Bacteroides_coprocola, and Alistipes_putredinis, among others. Fig 4 illustrates the corresponding importance scores and contribution rates of these 8 shared features among the top 20 features in the CRFS preprocessing method. Ultimately, these 8 features are incorporated into the model, indicating their significant role in the prediction process.

## 3.3 PGPM classifier performance analysis

**3.3.1 Evaluation of the performance of different models.** In this study, ACC stands for accuracy, which refers to the proportion of correctly classified instances out of the total number of instances when using the test set to evaluate a model in classification tasks. However, ACC has certain limitations and may not fully reflect the performance of a model. For example, it does not consider situations of class imbalance, where one class has significantly more samples than others. As a result, the model's performance cannot be fully assessed, leading to the introduction of the Area Under the Curve (AUC) and the ROC curve. The term ncorrect represents the number of correctly classified records, while ntotal represents the total number of test data. The calculation formulas are shown as follows in Eqs (3–1) to (3–2):

$$ACC = \frac{n_{correct}}{n_{total}} \tag{3-1}$$

$$TPRate = \frac{TP}{TP + FN} \tag{3-2}$$

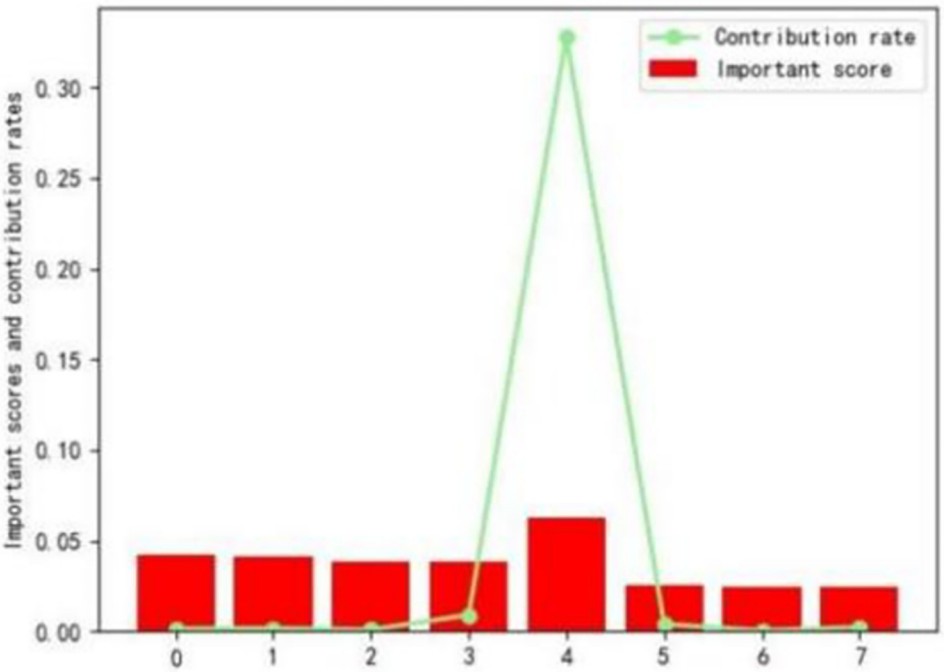

**Fig 4. CRFS significant scores and contributions.**

$$FPRate = \frac{FP}{TP + TN} \qquad (3-3)$$

Among them, True Positives (TP) refer to positive samples correctly predicted as positive, representing the number of positive instances correctly predicted; False Positives (FP) refer to negative samples incorrectly predicted as positive, representing the number of negative instances incorrectly predicted; True Negatives (TN) refer to negative samples correctly predicted as negative, representing the number of negative instances correctly predicted; False Negatives (FN) refer to positive samples incorrectly predicted as negative, representing the number of positive instances incorrectly predicted.

To compare the effectiveness of the PGPM proposed in this study with that of other commonly used neural networks for processing gut microbiota data, training and testing were conducted on this dataset, and the results are presented in Table 2. Table 2, shows that the classification performance of the PGPM method overall surpasses that of other commonly used classification models. For a more intuitive comparison of the differences in Mean Acc,

**Table 2. Experimental results of different models.**

| methodologies | Mean Acc | AUC | ROC |
|---|---|---|---|
| PGPM | 0.85 | 0.92 | 0.92 |
| DNN | 0.55 | 0.73 | 0.73 |
| LSTM | 0.50 | 0.58 | 0.58 |
| CNN | 0.53 | 0.60 | 0.60 |
| SVM | 0.78 | 0.88 | 0.88 |

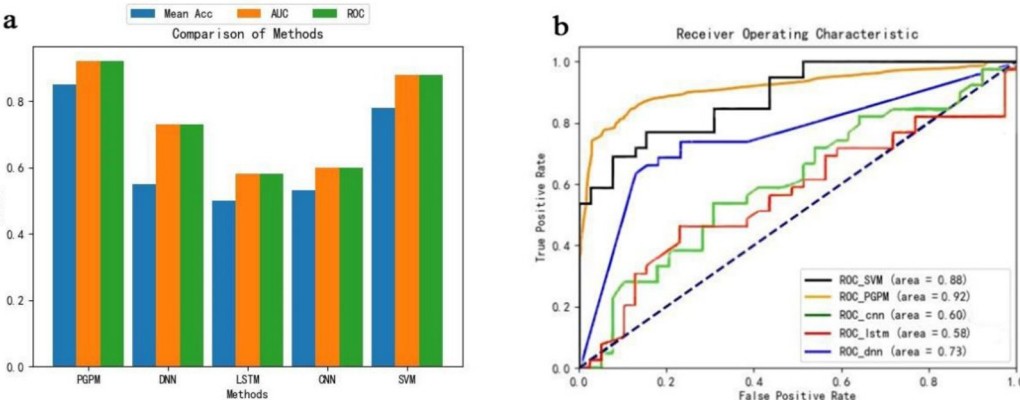

**Fig 5. Histograms of the different models and ROC curves of the different models.**

AUC, and ROC among the various models, this study provides bar graphs of the three indicators, as shown in Fig 5A. The ROC curve plot for the PGPM method is illustrated in Fig 5B.

Fig 5A, clearly shows that the PGPM exhibits significant advantages in Mean Acc, ROC, and AUC, and the comprehensive performance across all three indicators is notably high.

**3.3.2 PGPM ablation experiments.** In this experiment, to assess the individual impact of each module on the model's predictive ability, ablation experiments were conducted, as shown in Table 3. By comparing these experiments, we can observe the effects of different modules on the experimental results. The baseline was set as the LSTM model.

Based on the experimental list in Table 1, the corresponding model structures are constructed using the same hyperparameters, experiments are conducted using the same dataset, and the experimental results of the five methods are compared, as shown in Table 4. The comparison line graph is depicted in Fig 6.

From Table 4 and the line graph in Fig 6, it can be observed that as the methods continue to improve, the experimental results also show consistent enhancement. Comparing the results between Experiment 1 and Experiment 2, as well as between Experiment 1 and Experiment 3, it is evident that incorporating a single feature selection method improves the LSTM model's classification performance in terms of mean accuracy, AUC, and ROC. This suggests that feature simplification can reduce model complexity and enhance model performance to a certain extent.

Comparing Experiment 2, Experiment 3, and Experiment 4, it is apparent that the performance of the CRFS module surpasses that of a single feature selection method. Contrasting Experiment 1 with Experiment 5, it is clear that all metrics have improved. By combining the LSTM and SVM classification methods, the model's performance is further boosted.

Comparing Experiment 2, Experiment 3, Experiment 4, Experiment 5, and Experiment 6, it becomes evident that the contributions of the CRFS module and the PGPM method to the

**Table 3. List of ablation experiments.**

| serial number | Description of the experiment |
| --- | --- |
| 1 | baseline (in geodetic survey) |
| 2 | Addition of PCA module to the baseline |
| 3 | Adding RF modules to the baseline |
| 4 | Inclusion of the CRFS module in the baseline |
| 5 | Adding SVM modules to the baseline |
| 6 | PGPM |

**Table 4. Experimental results.**

| Experiment number | Mean Acc | AUC | ROC |
|:---:|:---:|:---:|:---:|
| 1 | 0.50 | 0.58 | 0.58 |
| 2 | 0.64 | 0.58 | 0.58 |
| 3 | 0.50 | 0.53 | 0.53 |
| 4 | 0.70 | 0.74 | 0.74 |
| 5 | 0.80 | 0.86 | 0.86 |
| 6 | 0.87 | 0.92 | 0.92 |

model's improvement exceed those of the individual methods, leading to superior overall performance. The experimental results demonstrate that the effectiveness of the PGPM surpasses previous research efforts. The experimental results consistently prove that the PGPM method is more effective than the methods used in previous studies.

## 4 Discussion

The gut microbiota played a crucial role in predicting Parkinson's disease [19]. Previous studies had clearly indicated the close relationship between the gut microbiota and Parkinson's disease. For example, the study by Bedarf et al. [20] found significant differences in the gut microbiota composition of Parkinson's disease patients compared to healthy controls. These differences were mainly reflected in the abundance changes of specific microorganisms, which might reveal particular pathophysiological processes of Parkinson's disease, providing new clues for its diagnosis and prediction. The core objective of this study was to develop an efficient and accurate prediction method for the early diagnosis of Parkinson's disease through in-depth analysis of gut microbiota data. The close relationship between the gut microbiota and Parkinson's disease has been widely studied. We propose a differential gut microbiota-based Parkinson's prediction method (PGPM) based on deep learning, aiming to capture the subtle differences in the gut microbiome that traditional machine learning [21] methods might miss, offering new perspectives and tools for Parkinson's disease prediction.

In this study, we explored different methods for predicting Parkinson's patients' performance. Compared to traditional methods (including DNN, LSTM, CNN, and SVM), our proposed method performed better, demonstrating its high capability in Parkinson's prediction classification. Precision, AUC, and ROC values were selected as key indicators to evaluate method performance, and the research results showed that the PGPM method achieved the best performance. In the comparison of classification performance after feature selection, it was found that feature dimensionality reduction could simplify the model complexity and improve model performance to a certain extent. Additionally, the combination of preprocessing methods led to more significant improvements in classification performance.

Our PGPM method achieved significant results in the classification prediction of Parkinson's disease, with a mean accuracy (Mean ACC) of 0.85, and both the area under the curve (AUC) and receiver operating characteristic curve (ROC) reaching 0.92. These results indicated that by deeply analyzing gut microbiota data, we could accurately distinguish Parkinson's disease patients from healthy individuals, providing strong support for the early diagnosis of Parkinson's disease.

Given the high-dimensional feature space and high redundancy of medical data, feature selection was necessary in data analysis. In this study, using the CRFS preprocessing method, eight gut microbiota features were selected, resulting in higher prediction accuracy for subsequent classification, with an increase of about 0.2 compared to single feature selection methods. This demonstrated the importance of feature selection for disease prediction. A study on

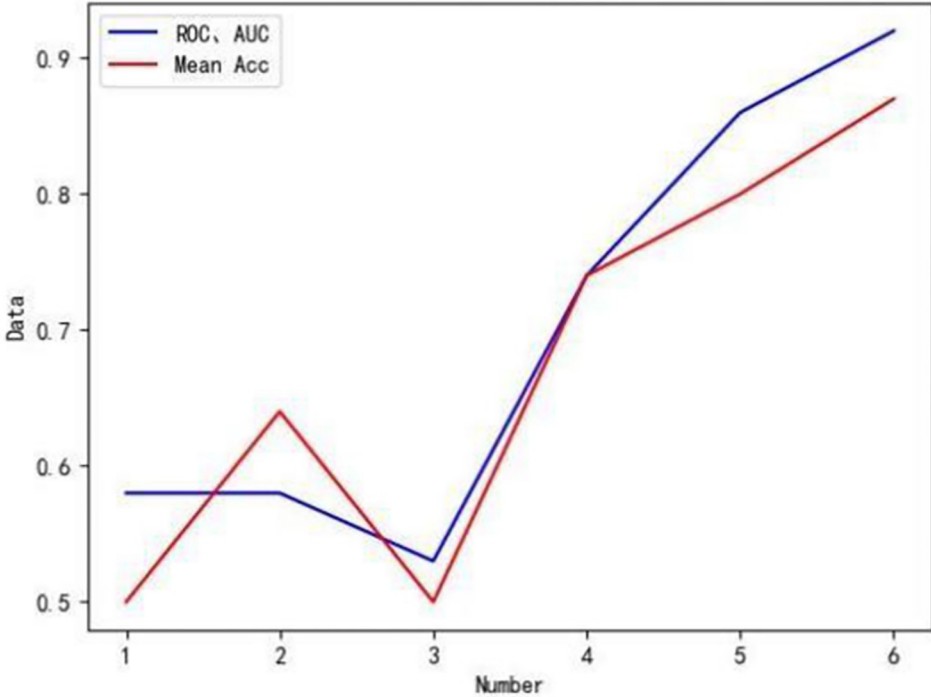

**Fig 6. Folded line comparison chart.**

the gut microbiota of diabetic patients also confirmed this, showing that selected gut microbiota features were crucial for the predictive ability of the model [22].

Furthermore, like other classifiers such as DNN, LSTM, CNN, and SVM, when dealing with high-dimensional feature spaces, redundant features, noisy features, and class imbalance in the data posed challenges to classification performance. Therefore, in this study, we combined LSTM with SVM, which improved the accuracy by about 0.3 compared to other methods (DNN, LSTM, CNN, and SVM). The experimental results also fully demonstrated the effectiveness of combining feature dimensionality reduction and combined classification models.

Compared to some methods developed for the microbiome in recent years, our method was simple, robust, and effective. Despite the significant achievements of this study, we acknowledged certain limitations. Future work would focus on expanding the sample size, improving result stability, and validating the external applicability and generalizability of this prediction model in larger independent validation groups. Additionally, we would further explore the deep relationship between gut microbiota and Parkinson's disease to achieve broader applications in personalized medicine.

In conclusion, the Parkinson's disease prediction model established in this study had achieved significant results, revealing the potential association between gut microbiota and Parkinson's disease. These findings might provide new ideas and methods for the early diagnosis and treatment of Parkinson's disease. Further research could deepen the understanding of the relationship between gut microbiota and Parkinson's disease and explore its potential in personalized medicine.

## 5 Conclusion

Support Vector Machine (SVM) and other machine learning methods are mainstream approaches for processing various gut microbiota data. In addition to the large volume of data,

there are many implicit correlations among the data. Moreover, the complex background of gut microbiota data makes it challenging for traditional machine learning and LSTM to obtain accurate features. Furthermore, there is a lack of research on the use of deep learning for classification prediction using gut microbiota data. Therefore, the PGPM method includes a complete set of methods ranging from feature selection to classification prediction. It accurately selects relevant features through the preprocessing process and utilizes a classification strategy combining LSTM-SVM to accomplish the classification prediction task. Overall, PGPM outperforms existing models and can effectively classify and predict Parkinson's gut microbiota. In future research, efforts will continue to accurately capture relevant features and focus on more precise classification model predictions. Additionally, this method can be extended to predict other diseases related to the gut microbiota.

## Supporting information

**S1 File.**
(RAR)

## Author Contributions

**Methodology:** Bo Yu.

**Supervision:** Min Zhang.

**Validation:** Hang Zhang.

**Writing – original draft:** Hang Zhang.

**Writing – review & editing:** Bo Yu, Min Zhang.

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
