## [Decision Letter · Decision Letter 0]

8 May 2024

PONE-D-24-12205Method for Predicting Parkinson's Disease through Differential Gut MicrobiotaPLOS ONE

Dear Dr. 张,

Thank you for submitting your manuscript to PLOS ONE. After careful consideration, we feel that it has merit but does not fully meet PLOS ONE’s publication criteria as it currently stands. Therefore, we invite you to submit a revised version of the manuscript that addresses the points raised during the review process.

**ACADEMIC EDITOR:**

Please change this manuscript according the reviewer's recommendation. 

We look forward to receiving your revised manuscript.

Kind regards,

Rocco Franco

Academic Editor

PLOS ONE

Additional Editor Comments:

Please adjust this paper according the reviewer's recommendation.

Regards

Reviewers' comments:

Reviewer's Responses to Questions

**Comments to the Author**

1. Is the manuscript technically sound, and do the data support the conclusions?

Reviewer #1: Yes

Reviewer #2: Yes

2. Has the statistical analysis been performed appropriately and rigorously? 

Reviewer #1: I Don't Know

Reviewer #2: N/A

3. Have the authors made all data underlying the findings in their manuscript fully available?

Reviewer #1: Yes

Reviewer #2: Yes

4. Is the manuscript presented in an intelligible fashion and written in standard English?

Reviewer #1: Yes

Reviewer #2: Yes

5. Review Comments to the Author

Reviewer #1: The study has Good design and a novel study but the study involves human specimens and requires an Ethical clearance. Authors have not discussed the results of their study in the "Discussion" section properly. They need to elaborate the discussion to compare and focus their results.

Reviewer #2: Thank you for allowing me to review the manuscript titled: Method for Predicting Parkinson's Disease through Differential Gut Microbiota. The manuscript is well-written, and the research work is of high quality.

I have a few comments:

1. Title: The current title needs to be less generalized and should provide readers with an idea of the actual content of the manuscript. As the study focuses on developing a Deep learning tool, it is advised to add some words related to Deep learning in the title to make it more interesting and specific

2. Materials and methods:

a)Secton 2.3 “In that study, a method for predicting Parkinson's disease patients using differential microbiota was implemented.” : Please quote the study and cite the reference

b)“The overall framework of the PGPM method constructed in that article was illustrated in Figure 1,”: Please quote the study and cite the reference

c)Section 2.5 .“ LSIM In that study, the classifier for the MDOS method was based on a combined classification strategy of LSTM-SVM” : Please quote the study and cite the reference

d)Provide details about who performed the annotation of deep learning tool.

e)How was blinding performed to avoid bias?

f)“ This experiment conducted comparative trials across multiple models, with identical hyperparameter settings as follows: the number of training epochs was set to 300, the initial learning rate was set at 0.001, the batch size was set to 6,” : please cite the reference and

g)“optimization algorithm used was Adaptive Moment Estimation (Adam)” : Provide details of company and Version of the software

3. Discussion: “ Bedarf et al. reported found significant differences in the composition of the gut microbiota between Parkinson's disease patients and healthy controls” : please cite the reference

4. Discussion can be elaborated as its too short.

6. PLOS authors have the option to publish the peer review history of their article (what does this mean?). If published, this will include your full peer review and any attached files.

Reviewer #1: **Yes: **Sumir Gandhi

Reviewer #2: No

---

## [Author Response · Author response to Decision Letter 0]

31 May 2024

Response to Reviewer 1:

Dear Sumir Gandhi:

 We greatly appreciate your positive feedback on our research and the issues you pointed out in this systematic review and meta-analysis. Based on your suggestions, we have made the relevant corrections in the document and have addressed your comments：

Comment1：The study has Good design and a novel study but the study involves human specimens and requires an Ethical clearance. Authors have not discussed the results of their study in the "Discussion" section properly. They need to elaborate the discussion to compare and focus their results.

Response: We greatly appreciate you pointing out this issue. This study, which involves human participants, has been reviewed and approved by the Ethics Committee of Xiangyang First People's Hospital. Written informed consent was obtained from the patients/participants for their participation in this study. Additionally, we have made the revisions you suggested to the discussion section of the manuscript and marked these changes accordingly.

Dear Reviewer 2:

Dear Reviewer:

 We greatly appreciate your positive feedback on our study and the issues you pointed out in this systematic review and meta-analysis. Based on your suggestions, we have made the necessary corrections in the document and have addressed your comments accordingly.

Comment 1：The current title needs to be less generalized and should provide readers with an idea of the actual content of the manuscript. As the study focuses on developing a Deep learning tool, it is advised to add some words related to Deep learning in the title to make it more interesting and specific.

Response1：Thank you very much for pointing out this issue. We have revised the title of the study to include the term "deep learning." The revised title is " Deep Learning-based Differential Gut Flora for Prediction of Parkinson's"

Comment a：Secton 2.3 “In that study, a method for predicting Parkinson's disease patients using differential microbiota was implemented.” : Please quote the study and cite the reference.

Response a:Thank you very much for pointing out this issue, and we apologize for the confusion. Due to our writing error, "In this study" was mistakenly written as "In that study," causing misunderstanding. The method mentioned is the one used in our current study, and there is no external reference to cite.

Comment b:“The overall framework of the PGPM method constructed in that article was illustrated in Figure 1,”: Please quote the study and cite the reference.

Response b: Thank you very much for pointing out this issue. It may have been a mistake in our writing that caused the misunderstanding. The PGPM method is the name of the method used in our current study, and it cannot be referenced to external literature.

Comment c：Section 2.5 .“ LSIM In that study, the classifier for the MDOS method was based on a combined classification strategy of LSTM-SVM” : Please quote the study and cite the reference.

Response c: Thank you very much for pointing out this issue. We apologize for the confusion caused by our writing error where "MDOS" was mistakenly used. It has been corrected to "PGPM" method, and therefore, there is no reference to cite. We sincerely apologize for any misunderstanding caused.

Comment d: Provide details about who performed the annotation of deep learning tool.

Response d: Thank you very much for pointing out the issue. This study was implemented in Python (version 3.9.12) using publicly available standard libraries, including pandas (version 1.5.2), numpy (version 1.22.4), scikit-learn (version 1.2.0), torch (version 1.12), and matplotlib (version 3.6.2). We have also added this information to the manuscript.

Comment e: How was blinding performed to avoid bias?

Response e: Thank you very much for pointing out this issue. In this study, the data were collected from Parkinson's patients and their healthy spouses in the same region, with a gender ratio of 1:1. During the experiments, we utilized the DataLoader method to randomly shuffle the sample order in the dataset at the beginning of each epoch to avoid bias.

Comment f: This experiment conducted comparative trials across multiple models, with identical hyperparameter settings as follows: the number of training epochs was set to 300, the initial learning rate was set at 0.001, the batch size was set to 6,” : please cite the reference.

Response f: Thank you very much for pointing out this issue. We have added references to the manuscript as suggested. However, some of the parameters are based on our experimental experience and thus cannot be referenced.

Comment g:“optimization algorithm used was Adaptive Moment Estimation (Adam)” : Provide details of company and Version of the software.

Response g:Thank you very much for pointing out this issue. The GPU used for training in this study is the NVIDIA GeForce GTX1060 laptop GPU, with 16GB of memory and 1280 CUDA cores. We have already added this information to the manuscript.

Comment 3：Discussion: “ Bedarf et al. reported found significant differences in the composition of the gut microbiota between Parkinson's disease patients and healthy controls” : please cite the reference.

Response 3:Thank you very much for pointing out the issue. We have now added the citation of the reference at this location.

Comment 4：Discussion can be elaborated as its too short.

Comment 4：Thank you very much for bringing up this issue. We have addressed the problem of the "Discussion" section being too brief as you pointed out, and we have made the necessary revisions, which are now indicated in the manuscript.

---

## [Decision Letter · Decision Letter 1]

26 Jun 2024

PONE-D-24-12205R1Deep Learning-based Differential Gut Flora for Prediction of Parkinson'sPLOS ONE

Dear Dr. 张,

Thank you for submitting your manuscript to PLOS ONE. After careful consideration, we feel that it has merit but does not fully meet PLOS ONE’s publication criteria as it currently stands. Therefore, we invite you to submit a revised version of the manuscript that addresses the points raised during the review process. 

We look forward to receiving your revised manuscript.

Kind regards,

Rocco Franco

Academic Editor

PLOS ONE

Journal Requirements:

Additional Editor Comments:

Dear Authors,

Please follow the reviewer recommendation

Regards

Reviewers' comments:

Reviewer's Responses to Questions

**Comments to the Author**

1. If the authors have adequately addressed your comments raised in a previous round of review and you feel that this manuscript is now acceptable for publication, you may indicate that here to bypass the “Comments to the Author” section, enter your conflict of interest statement in the “Confidential to Editor” section, and submit your "Accept" recommendation.

Reviewer #1: All comments have been addressed

Reviewer #2: All comments have been addressed

2. Is the manuscript technically sound, and do the data support the conclusions?

Reviewer #1: Yes

Reviewer #2: Yes

3. Has the statistical analysis been performed appropriately and rigorously? 

Reviewer #1: I Don't Know

Reviewer #2: Yes

4. Have the authors made all data underlying the findings in their manuscript fully available?

Reviewer #1: Yes

Reviewer #2: Yes

5. Is the manuscript presented in an intelligible fashion and written in standard English?

Reviewer #1: Yes

Reviewer #2: Yes

6. Review Comments to the Author

Reviewer #1: The authors have made certain changes in the manuscript. But cupola of points still need to be addressed--

1. Specify aims and objective of the study.

2. I discussion section, authors have focussed in the predictive model but not on their results. Please elaborate on that.

Reviewer #2: All comments were answered well and necessary change were made in the manuscript. I believe the manuscript can be accepted in the present from.

7. PLOS authors have the option to publish the peer review history of their article (what does this mean?). If published, this will include your full peer review and any attached files.

Reviewer #1: **Yes: **Sumir Gandhi

Reviewer #2: No

---

## [Author Response · Author response to Decision Letter 1]

8 Jul 2024

Response to Reviewer 1:

Dear Professor Sumir Gandhi:

 We greatly appreciate your positive feedback on our research and the issues you pointed out in this systematic review and meta-analysis. Based on your suggestions, we have made the relevant corrections in the document and have addressed your comments：

Comment1：Specify aims and objective of the study.

Response1: We greatly appreciate you pointing out this issue. The core objective of this study is to develop an efficient and accurate prediction method for the early diagnosis of Parkinson's disease through an in-depth analysis of gut microbiota data. We propose a differential gut microbiota-based Parkinson's prediction method (PGPM) based on deep learning, aiming to capture the subtle differences in the gut microbiome that traditional statistical methods may miss, offering new perspectives and tools for Parkinson's disease prediction. Additionally, we have incorporated the study's aims and objectives into the abstract and introduction sections and highlighted them in red. Thank you again for pointing out this issue.

Comment2：I discussion section, authors have focussed in the predictive model but not on their results. Please elaborate on that.

Response2: We greatly appreciate you pointing out this issue. We have realized that the discussion section was overly focused on our model while neglecting our results. We have made revisions to the original text, adding a section on the results of our study to the discussion. Thank you again for your guidance on our research.

Response to Reviewer 2:

Dear reviewers2：

 It is a great honour to have your approval of this work.

---

## [Editor Report · Acceptance letter]

1 Sep 2024

PONE-D-24-12205R2 

PLOS ONE

Dear Dr. Zhang, 

I'm pleased to inform you that your manuscript has been deemed suitable for publication in PLOS ONE. Congratulations! Your manuscript is now being handed over to our production team.

Kind regards, 

on behalf of

Professor Upaka Rathnayake 

Academic Editor

PLOS ONE